# A Pleiotropic and Functionally Divergent *RAC3* Variant Disrupts Neurodevelopment and Impacts Organogenesis

**DOI:** 10.3390/cells14191499

**Published:** 2025-09-24

**Authors:** Ryota Sugawara, Marcello Scala, Sara Cabet, Carine Abel, Louis Januel, Gaetan Lesca, Laurent Guibaud, Frédérique Le Breton, Hiroshi Ueda, Hidenori Tabata, Hidenori Ito, Koh-ichi Nagata

**Affiliations:** 1Department of Molecular Neurobiology, Institute for Developmental Research, Aichi Developmental Disability Center, 713-8 Kamiya, Kasugai 480-0392, Japan; suga.ryo2018@gmail.com (R.S.); tabata@inst-hsc.jp (H.T.); itohide@inst-hsc.jp (H.I.); 2Department of Neurosciences, Rehabilitation, Ophthalmology, Genetics, Maternal and Child Health, University of Genoa, 16132 Genoa, Italy; marcelloscala87@gmail.com; 3Unit of Medical Genetics, IRCCS Giannina Gaslini Institute, 16147 Genoa, Italy; 4Pediatric and Fetal Imaging Department, Femme-Mère-Enfant Hospital, Hospices Civils de Lyon, Claude Bernard Lyon 1 University, 69500 Lyon, France; sara.cabet@chu-lyon.fr (S.C.); laurent.guibaud@chu-lyon.fr (L.G.); 5Pathology and Genetics of Neuron and Muscle, Neuromyogene Institute, CNRS UMR 5261 INSERM U1315, University of Lyon–Université Claude Bernard Lyon 1, 69500 Lyon, France; gaetan.lesca@chu-lyon.fr; 6Department of Genetics, Groupement Hospitalier Est, Hospices Civils de Lyon, 69777 Bron, France; carine.abel@chu-lyon.fr (C.A.); louis.januel@chu-lyon.fr (L.J.); 7Plan France Medecine Génomique, Auragen Laboratory, 69003 Lyon, France; 8Department of Pathology, Hospices Civils de Lyon, 69777 Bron, France; frederique.le-breton@chu-lyon.fr; 9United Graduate School of Drug Discovery and Medical Information Sciences, Gifu University, Yanagido, Gifu 501-1193, Japan; ueda.hiroshi.j6@f.gifu-u.ac.jp; 10Department of Chemistry and Biomolecular Science, Faculty of Engineering, Gifu University, Yanagido, Gifu 501-1193, Japan; 11Center for One Medicine Innovative Translational Research (COMIT), Gifu University, Yanagido, Gifu 501-1193, Japan; 12Department of Neurochemistry, Nagoya University Graduate School of Medicine, 65 Tsurumai-cho, Nagoya 466-8550, Japan

**Keywords:** *RAC3*, small GTPase, neurodevelopmental disorder, fetopathy, functional divergence

## Abstract

*RAC3* encodes a small Rho-family GTPase essential for cytoskeletal regulation and neurodevelopment, and de novo *RAC3* variants typically act as gain-of-function alleles that cause severe neurodevelopmental disorders. In this study, we analyzed a fetus with multisystem congenital anomalies and identified a de novo RAC3 p.(T17R) variant by genome sequencing. To elucidate the pathogenicity of this variant, we combined in silico variant prioritization, structural and energetic modeling, and pathogenicity prediction with in vitro biochemical assays, including GDP/GTP exchange, GTP hydrolysis, effector pull-down, and luciferase reporter analyses in COS7 cells, as well as morphological analysis of primary hippocampal neurons. Furthermore, we performed in vivo analyses using a mouse in utero electroporation to assess cortical neuron migration, axon extension, and dendritic development. Our biochemical results suggest that RAC3-T17R exhibits markedly increased GDP/GTP exchange, with a preference for GDP binding, and undetectable GTP hydrolysis. The mutant displayed minimal binding to canonical RAC effectors (PAK1, MLK2, and N-WASP) and failed to activate SRF-, NFκB-, or AP1-dependent transcription. Neuronal overexpression of RAC3-T17R impaired axon formation in vitro, while in vivo expression delayed cortical neuron migration and axon extension and reduced dendritic arborization. Clinically, the fetus exhibited corpus callosum agenesis, microcephaly, organomegaly, and limb contractures. Collectively, these findings indicate that the RAC3 p.(T17R) variant may represent a signaling-deficient allele with pleiotropic, variant-specific mechanisms that disrupt corticogenesis and broader organogenesis. Our multi-tiered in silico–in vitro–in vivo approach demonstrates that noncanonical *RAC3* variants can produce complex, multisystem developmental phenotypes beyond previously recognized *RAC3*-related neurodevelopmental disorders.

## 1. Introduction

The *RAC3* gene encodes a member of the Rho family small GTPases, which regulate intracellular signaling and actin cytoskeletal organization, significantly contributing to neuronal development and function [1,2]. Unlike its closely related homologs, RAC1 and RAC2, which are ubiquitously expressed or restricted to hematopoietic cells, respectively, *RAC3* is predominantly expressed in the brain, where it is crucial for neuronal migration, neurite outgrowth, dendritic branching, and synaptic plasticity [3]. These functions are tightly regulated by the GDP/GTP cycle, which is facilitated by guanine nucleotide exchange factors (GEFs) and GTPase-activating proteins (GAPs) [4]. Given its essential role in neural development, disruption of RAC3 function by pathogenic variants can lead to aberrant neuronal migration, defective dendritic branching, and impaired synaptic plasticity. In fact, de novo *RAC3* variants have been implicated in a severe neurodevelopmental disorder (NDD) known as NEDBAF (Neurodevelopmental Disorder with Brain Anomalies and Dysmorphic Facies, MIM #618577) [5,6,7,8,9,10].

Deleterious disease-related *RAC3* variants disrupt critical domains involved in GDP/GTP binding, GTP hydrolysis, and effector interactions, leading to altered cellular signaling and abnormal brain development [8,11,12,13]. Most substitutions cluster in highly conserved functional regions, such as Switch I/II and G1-G5 boxes. The Switch I/II regions mediate interactions with GEFs, GAPs, and effectors, while the G1-G5 boxes maintain the structural integrity of the nucleotide-binding pocket and are critical for GDP/GTP exchange and GTP hydrolysis [6,7,8,9,11]. A notable exception is the c.276T>A, p.N92K variant, which is located outside these core functional regions and was identified in a prenatal case featuring complex brain malformations [10]. Despite its atypical location, the p.N92K mutant functions as an activated form due to its resistance to GAP-mediated inactivation [13]. Collectively, all *RAC3* variants reported to date exhibit gain-of-function properties, leading to abnormal cortical neuron migration, impaired axon elongation, and dysregulated PAK1-mediated signaling pathways [8,11,12,13].

Here, we report a severe fetopathy, featuring brain malformations and extra-neurological anomalies, in a fetus harboring a novel de novo missense change in *RAC3*. The p.(T17R) variant is the second *RAC3* substitution known to affect the G1 box. Through complementary lines of biochemical and molecular investigation, we provide evidence that this variant exerts distinctive pathogenic mechanisms compared to classic *RAC3* substitutions, underlying a highly atypical clinical presentation. These findings offer insights into the disease mechanisms of *RAC3* variants beyond NDDs, involving RAC3 dysfunction in new and complex human phenotypes.

## 2. Materials and Methods

### 2.1. Genome Sequencing and Variant Analysis

Genome sequencing was performed as part of France Genomic Medicine Plan. Whole blood extracted genomic DNA was sequenced according to standard procedures for a PCR-Free genome on a NovaSeq6000 instrument (Illumina, San Diego, CA, USA). Sequencing data were aligned to the GRCh38.p13 full assembly using Burrows-Wheeler Aligner (v0.7). Variants were called by several algorithms including Genome Analysis Toolkit (GATK v4), BCFtools (v1.10), Manta (v1.6.0), and CNVnator (v0.4), and annotated using the variant effect predictor (VEP) [14]. Detected variants were prioritized using in-house procedures. Further details are available upon request at http://www.auragen.fr (accessed on 1 July 2025). Variants were filtered according to minor allele frequency ≤ 0.001 in Genome Aggregation Database (gnomAD v4.1.0), conservation of affected residues (Genomic Evolutionary Rate Profiling—GERP), and predicted functional impact through different in silico tools [15,16]. Candidate variants were then classified according to the ACMG-AMP guidelines [17]. The candidate *RAC3* variant is reported according to the NM_005052.3 transcript (NP_005043.1) (https://www.ncbi.nlm.nih.gov/nuccore/NM_005052.3, accessed on 1 July 2025) and has been submitted to the LOVD database with the following accession number: #0001045539. The presence of copy number variations (CNVs) was excluded through sequencing-based CNV calling. No additional candidate variants in other morbid genes were identified.

### 2.2. Plasmids

Site-directed mutagenesis was performed using the KOD-Plus Mutagenesis Kit (Toyobo Inc., Osaka, Japan, Cat# SMK-101) to generate RAC3-T17R and RAC3-Q61L, the latter being a constitutively active form associated with pathogenicity [7,8,18], with pCAG-Myc-RAC3 serving as the template. Additionally, wild-type and mutant *RAC3* constructs were cloned into the pTriEx-4 vector (Merck, Darmstadt, Germany, Cat# 70824). For pull-down assays, pGS21a vectors (GenScript, Piscataway, NJ, Cat# SD0121) encoding the RAC-binding regions (RBRs) of key effectors were generated as previously reported [12]. The following RBRs were included: human PAK1 (p21-activated kinase 1; amino acids, aa 67–150), human MLK2 (mixed lineage kinase 2/MAPKKK10; aa 401–550), and rat N-WASP (Neural Wiskott-Aldrich syndrome protein; aa 191–270). For transcriptional analysis, luciferase reporter plasmids were obtained from Promega (Madison, WI, USA), including pGL4.74[hRluc/TK] (Cat# E6921, control reporter), pGL4.44[luc2P/AP1-RE/Hygro] (Cat# E4111, AP1-luciferase reporter), and pGL4.32[luc2P/NF-κB-RE/Hygro] (Cat# E8491, NFκB-luciferase reporter). The pSRF-Luc plasmid was obtained from Agilent Technologies (La Jolla, CA, USA, Cat# 219082). All plasmid constructs were verified by DNA sequencing to ensure accuracy.

### 2.3. Antibodies

The antibodies used were as follows: anti-GFP (Medical & Biological Laboratories, Nagoya, Japan, Cat# 598, RRID: AB_591819; Nacalai Tesque, Kyoto, Japan, Cat# 04404-84, RRID: AB_10013361), and anti-Myc (Medical & Biological Laboratories, Nagoya, Japan, Cat# M047-3, RRID: AB_591112). Alexa Fluor 488 goat anti-Rabbit IgG (Thermo Fisher Scientific, Waltham, MA, USA, Cat# A-11034, RRID:AB_2576217), and Alexa Fluor 568 goat anti-Mouse IgG (Thermo Fisher Scientific, Cat# A-11031, RRID:AB_144696) were used as secondary antibodies. DAPI (Nichirei Bioscience, Tokyo, Japan) was used for DNA staining.

### 2.4. GTP/GDP Exchange and GTP Hydrolysis Assays

His-tag-fused RAC3 and RAC3-T17R proteins were expressed and purified by use of Ni-NTA agarose (FUJIFILM Wako Pure Chemical Co., Osaka, Japan, Cat#141-09764) following the manufacturer’s instructions [19]. To evaluate the intrinsic GTP/GDP exchange activity, we monitored the release of methylanthraniloyl (mant)-GDP (Sigma-Aldrich, St Louis, MO, USA, Cat# 69244) using an established fluorescence-based method [20]. The GTP hydrolysis activity was assessed using the GTPase-Glo Assay Kit (Promega, Madison, WI, USA, Cat# V7681) in accordance with the manufacturer’s instructions [21].

### 2.5. Cell Culture and Transfection

COS7 cells (monkey kidney fibroblasts), which were obtained from American Type Culture Collection (RRID: CVCL_0224), and primary hippocampal neurons isolated from embryonic day (E) 16.5 mice were cultured as previously described [22]. Transient transfection into COS7 was performed using the polyethyleneimine “MAX” reagent (Polysciences Inc., Warrington, PA, USA, Cat# 24765), while neuronal transfection was conducted using the Neon transfection system (Thermo Fisher Scientific).

### 2.6. Pull-Down Assay

COS7 cells were transfected with Myc-tagged *RAC3* constructs (WT or T17R) and cultured for 48 h. Cells were lysed in ice-cold lysis buffer (50 mM Tris-HCl pH 7.5, 150 mM NaCl, 5 mM MgCl_2_, 0.1% SDS, 1% Nonidet P-40, 0.5% deoxycholate, 1 mM Na_3_VO_4_ and 1/100 vol. of protease inhibitor cocktail) and clarified by centrifugation at 13,000× *g* for 10 min. Recombinant GST-tagged RAC-binding regions of PAK1, MLK2, and N-WASP were immobilized on glutathione-Sepharose 4B beads (Cytiva, Uppsala, Sweden, Cat# 17075605) and incubated with the lysates at 4 °C for 1 h with gentle rotation. Beads were washed three times with lysis buffer and bound proteins were eluted by boiling in SDS sample buffer. For Western blotting, total cell lysates (input) and pull-down fractions were separated by SDS-PAGE (15% polyacrylamide gels) and transferred onto nitrocellulose membranes (Supported Nitrocellulose; GVS, Bologna, Italy, Cat# 1212590). Immunoreactive bands were detected using Western Lightning Plus ECL (Revvity Inc., Waltham, MA, USA, Cat# NEL 104001EA) and visualized on LAS-4000 luminescent image analyzer (GE Healthcare Life Sciences, Chicago, IL, USA)

### 2.7. In Utero Electroporation and Analysis of Cortical Neuron Migration

In utero electroporation was performed following established protocols [23,24]. Timed- pregnant ICR mice (Japan SLC, Shizuoka, Japan) were deeply anesthetized at E14 using a cocktail of medetomidine (0.75 mg/kg), midazolam (4 mg/kg), and butorphanol (5 mg/kg) [25]. pCAG-Myc (control), pCAG-Myc-RAC3, or pCAG-Myc-RAC3-T17R (0.1 μg each), and pCAG-EGFP (0.5 μg), were microinjected into the lateral ventricle of embryonic brains using a glass micropipette. Electroporation was conducted using an electroporator (NEPA21, NEPA Gene, Chiba, Japan), delivering five electrical pulses (50 ms, 35 V) at 450 ms intervals. This approach allows targeted plasmid introduction into the somatosensory region of the parietal lobe. E14 neuronal progenitor cells in the VZ are destined to become layer II/III pyramidal neurons in the adult stage. The brains were then collected at specific postnatal days (P0 or P7 for the analysis of neuronal migration and axon elongation, and P7 for dendritic development), followed by fixation, sectioning, and analysis. For quantitative analysis of GFP-positive cell distribution, coronal sections were divided into three bins, and the number of labeled cells in each bin was counted. A minimum of three slices per brain were analyzed. All experiments were conducted during the daytime, and no animals were excluded or died during the study.

All mice were housed under specific-pathogen-free (SPF) conditions with a 12 h light/dark cycle, controlled temperature (22 ± 1 °C) and humidity (55 ± 10%), and ad libitum access to food and water. A total of 18 embryos from 6 pregnant females were analyzed in the current study (see figure legends for sample sizes). Unless otherwise noted, both male and female embryos were included. All procedures were performed in accordance with the ARRIVE guidelines.

### 2.8. Immunofluorescence

Immunofluorescence was performed as described previously [11]. Images of cultured cells were acquired using a BZ-9000 fluorescence microscope (Keyence, Osaka, Japan) or an LSM-880 confocal laser scanning microscope (Carl Zeiss, Oberkochen, Germany). For cortical tissue slices, brains were embedded in 3% agarose, sectioned into 100 μm slices using a vibratome, and imaged with an LSM-880. The acquired images were analyzed using ImageJ software (version 2.9.0; US National Institutes of Health, Bethesda, MD; https://imagej.nih.gov/, accessed on 1 July 2025) to assess cell morphology and fluorescence intensity.

### 2.9. Estimation of Dendritic Arborization and Axon Growth In Vivo

In utero electroporation was carried out as described earlier [8]. To measure the length and number of branching points of dendrites in post-migratory (mature) neurons in layer II-III of the cerebral cortex, images of GFP-positive cells at P7 were acquired using a Zeiss880 confocal microscope. ImageJ software was used for the quantitative analyses of dendritic length and Sholl test. To evaluate axonal elongation, GFP signal intensity of the callosal axons was measured at P0 or P7 using imageJ software in distinct regions (bin 1–4). The relative intensities of bins were normalized with bin 1 as 1.0, and compared using R software (version 4.5.1; https://www.R-project.org/, accessed on 1 July 2025).

## 3. Results

### 3.1. Case Report

This case involves an aborted female fetus conceived by non-consanguineous, healthy parents of European ancestry (French). Family history was negative for NDDs or other syndromes associated with multiple congenital anomalies. Pregnancy course was uneventful until severe brain abnormalities were identified on prenatal brain MRI. At 23 weeks of gestation, MRI imaging revealed significant supratentorial and infratentorial malformations (Figure 1A). These abnormalities included reduced white matter volume, corpus callosum agenesis, and abnormal cerebral ventricles development. Severe pontocerebellar hypoplasia was also observed. Based on these findings, the parents opted for pregnancy termination.

### 3.2. Post-Mortem Pathology

Post-mortem examination confirmed primary microcephaly and significant dysmorphic features (Figure 1B). Of note, the fetus also showed significant organomegaly affecting multiple organs, including lungs, heart, kidneys, adrenal glands, and spleen. Additional relevant findings were bilateral hand contractures with congenital anomalies involving all fingers (Figure 1C). Neuropathological assessment of the fetal brain confirmed the malformations previously identified on prenatal MRI (Figure 2A). A more detailed dissection of brain defects revealed the presence of an abnormally sharp distinction between grey and white matter (Figure 2B), suggesting an underlying neuronal migration abnormality. Furthermore, there was severe hypoplasia of the cerebellar vermis and pons (Figure 2C).

### 3.3. Variant Identification and In Silico Analysis

Trio-genome sequencing identified a de novo heterozygous variant in *RAC3* in the fetus, NM_005052.3: c.50C>G p.(T17R) (Figure 3A). This variant was not present in gnomAD (V4.1.0) and affects a conserved residue of the protein (GERP score = 3.44), which is the last amino acid of the G1 domain of the protein (Figure 3B). The Thr17 residue is highly intolerant to variation (Figure 3C), and its substitution with Arg may affect protein stability and interactions with surrounding amino acids (Figure 3D). The p.(T17R) variant is predicted to be highly damaging by in silico tools (CADD score = 28; AlphaMissense score = 0.997; REVEL score = 0.899) and is classified as likely pathogenic according to the ACMG/AMP guidelines (Appendix A).

### 3.4. Biochemical Properties of RAC3-T17R

To elucidate the pathophysiological characteristics of RAC3-T17R, we evaluated the relative binding affinities of RAC3-T17R for GDP and GTP using a mantGDP dissociation assay. At the start of the assay, the relative fluorescence intensity of RAC3-T17R was comparable to that of wild-type RAC3 suggesting that RAC3-T17R does not favor a nucleotide-free state, unlike the dominant-negative RAC3-T17N. Upon addition of a non-hydrolysable GTP analog (GppNHp), fluorescence intensity decreased, with the reduction occurring significantly faster in RAC3-T17R than in wild-type RAC3 (Figure 4A,B). In the absence of GppNHp, RAC3-T17R still exhibited a more rapid decrease in fluorescence than wild-type RAC3, although the rate was slower than that observed with GTP (Figure 4A,B). We then performed the assay using GDP instead of GppNHp. Under these conditions, RAC3-T17R showed a much more pronounced decrease in fluorescence compared with wild-type RAC3, with a rate substantially faster than that observed in the presence of GppNHp (Figure 4C,D). These observations suggest that, while the accelerated decrease in RAC3-T17R fluorescence in the presence of GppNHp or GDP reflects enhanced GDP/GTP exchange activity, RAC3-T17R shows a preference for GDP over GTP. We also attempted to perform the GTP/GDP exchange assay for RAC3-T17N, a well-established dominant-negative variant, but were unable to do so because mantGDP could not be loaded—likely due to the mutant’s preference for a nucleotide-free conformation. Collectively, these findings suggest that RAC3-T17R biochemically behaves as an inactive, GDP-bound variant, distinct from other pathogenic RAC3 variants reported to date [8,11,12,13]. However, it cannot be excluded that RAC3-T17R may exist in a GTP-bound state under physiological conditions, in which GTP concentrations are substantially higher than GDP. In contrast, GTP hydrolysis activity of RAC3-T17R was undetectable, as luminescence levels remained unchanged under the assay conditions (Figure 4E). These results suggest that, if GDP is exchanged for GTP, RAC3-T17R may fail to hydrolyze GTP efficiently and thus remain trapped in a GTP-bound state.

### 3.5. Biological Properties of RAC3-T17R In Vitro

To elucidate the impact of the p.(T17R) variant on neuronal morphology, we analyzed primary cultured hippocampal neurons transfected with either pCAG-Myc (empty vector), pCAG-Myc-RAC3, or pCAG-Myc-RAC3-T17R. Neurons expressing RAC3 displayed normal differentiation, similar to those transfected with the empty vector, characterized by the extension of a single axon (Figure 5A–C). This suggests that basal RAC3 activity has minimal influence on neuronal differentiation in vitro. In contrast, neurons expressing RAC3-T17R failed to extend axons (Figure 5A–C). These findings indicate that RAC3-T17R disrupts neuronal morphology and differentiation. Notably, typical lamellipodia, which were evident in neurons expressing other activated RAC3 variants [8], were rarely observed in RAC3-T17R-expressing neurons (Figure 5A).

### 3.6. Interaction of RAC3-T17R with Downstream Effectors

We performed pull-down experiments to assess the impact of the p.(T17R) variant on downstream signaling pathways. Specifically, we conducted GST pull-down assays using COS7 cells expressing Myc-tagged RAC3 (wild type or T17R) and recombinant RBRs of PAK1, MLK2, and N-WASP, followed by Western blot analysis. The results demonstrated that RAC3-T17R displayed markedly weak binding to all three effector proteins, similar to wild-type RAC3 (Figure 6A–D). These results suggest that RAC3-T17R serves as an inactive, GDP-bound variant, although the possibility that it interacts with and activates as-yet-uncharacterized effector(s) cannot be excluded.

Next, based on the involvement of Rho family members in these pathways, we investigated gene expression pathways associated with SRF, NFκB, and AP1, using a transient expression method in COS7 cells [26,27,28]. Our findings revealed that RAC3-T17R had limited impact on the transcriptional activity of these genes compared to the wild type (Figure 6E–G). These results are consistent with the defective interaction of the mutant with the various downstream effectors tested, resulting in the reduced activation of SRF, NFκB, and AP1 signaling.

### 3.7. Effects of the p.(T17R) Variant on Neuronal Migration During Corticogenesis In Vivo

The post-mortem pathological assessment of the fetal brain revealed a markedly distinct contrast between the cortical grey matter and subcortical white matter, suggestive of a neuronal migration defect (Figure 2B). Hence, we employed in utero electroporation at E14 to introduce pCAG-EGFP together with either pCAG-Myc (empty vector), pCAG-Myc-RAC3 (WT), or pCAG-Myc-RAC3-T17R into progenitor cells in the ventricular zone (VZ). The distribution of GFP-labeled neurons was then analyzed at P0. As shown in Figure 7A,B, neurons expressing either the control vector or RAC3 successfully migrated to the upper regions of the cortical plate (CP), particularly to bin 1 corresponding to layers II/III. In contrast, a significant proportion of neurons expressing the RAC3-T17R variant remained in the VZ/subventricular zone (SVZ) (bin 3). It is worth noting that the efficiency of transfection was influenced by the surface area exposed to the ventricular lumen, where the plasmid DNA was delivered. As a result, neurons incorporating higher amounts of plasmid showed more severe variant-related effects. In this context, GFP expression levels were highest in bin 3, followed by bin 2 and bin 1, in descending order (Figure 7A: T17R panel). To examine the long-term effects of the variant, we also assessed neuronal positioning at P7. By this time, RAC3-T17R-expressing neurons had largely progressed toward their destined cortical layers but did not fully reach their appropriate laminar positions, indicating a statistically significant migration delay (Figure 7C,D). Comprehensively, these findings indicate that the p.(T17R) variant compromises the precise laminar targeting of cortical neurons.

### 3.8. Effects of the p.(T17R) Variant on Axon Growth During Cortical Development In Vivo

The presence of corpus callosum agenesis and microcephaly in the fetus suggest that the p.(T17R) substitution may impair axonal growth in vivo. To investigate this, we assessed axonal development of layer II/III pyramidal neurons in the parietal cortex during corticogenesis. At E14, we performed in utero electroporation to introduce pCAG-GFP along with either pCAG-Myc-RAC3 (WT) or pCAG-Myc-RAC3-T17R. At P0, visualization of the corpus callosum revealed delayed axonal extension in RAC3-T17R-expressing neurons compared to control cells (Figure 8A). Quantitative analyses confirmed this phenotype (Figure 8B). To assess long-term effects, we then examined axon growth at P7. At this stage, axon bundles of RAC3-T17R-expressing neurons had extended into the contralateral white matter, similar to control neurons (Figure 8C,D). These findings suggest that the p.(T17R) variant delays, but does not permanently impair, axonal growth during cortical development.

### 3.9. Effects of the p.(T17R) Variant on Dendritic Growth During Cortical Development In Vivo

Synaptic dysfunction is a relevant component in the pathophysiology of NDDs, leading us to hypothesize that the p.(T17R) substitution may contribute to the defective synaptic network formation. To explore this, we assessed the effect of the variant on the development of dendritic branching. When pCAG-Myc-RAC3-T17R was electroporated into the VZ cells at E14.5 and analyses were performed at P7, dendritic branch point number was reduced in the deficient neurons (Figure 9A,B). We also found that both basal and apical dendritic arbor development was suppressed (Figure 9C–E). We concluded that the p.(T17R) variant plays a crucial role in synaptic network formation through the regulation of dendritic development. As such, this substitution can lead to a synaptogenesis disruption, underlying significant neurodevelopmental defects.

Regarding statistical analysis, cell selection and tracing for all cell imaging experiments were performed in a blinded manner using ImageJ software. Statistical analysis was carried out using Prism 9 (GraphPad Software, Boston, MA, USA). The data were analyzed using the following tests: one-way ANOVA with Tukey’s post hoc test (Figure 4B,D, Figure 5B,C and Figure 6B–D); one-way ANOVA with Dunnett’s post hoc test (Figure 6 E–G); two-way ANOVA with Tukey’s post hoc test (Figure 7B,D), two-way ANOVA with Šídák’s post hoc test (Figure 8B,D and Figure 9B), and an unpaired *t* test (Figure 9C–D). A *p*-value of less than 0.05 was considered statistically significant.

## 4. Discussion

In this study, we have identified a novel and unique variant in *RAC3* in association with a severe fetopathy with multisystem congenital defects. Through several lines of functional investigation, we provided evidence that the de novo p.(T17R) variant exerts damaging effects on RAC3 function, leading to the disruption of crucial neurodevelopmental processes. We showed that this variant exhibits a functionally divergent profile compared to other RAC3 variants, resulting in distinct disease mechanisms. These findings not only expand the current spectrum of human phenotypes associated with RAC3 dysfunction to a new clinical condition, but also provides insights into previously uncharacterized pathophysiological mechanisms underlying *RAC3*-related disorders.

The vast majority of disease-causing *RAC3* reported to date are located within the Switch I/II regions or the G1–G5 boxes, that are critical for the GTPase function of RAC3 [6,7,8,9,10,11,12]. The only exception is the p.(N92K) variant, detected in a prenatal case of severe malformations of cortical development. This substitution affects a residue located outside these core functional regions but exerts highly disruptive effects on fundamental neurodevelopmental processes [13]. The p.(T17R) variant identified in our case affects the terminal residue of the G1 box, a highly conserved motif essential for GDP/GTP binding and GTP hydrolysis. Previously reported *RAC3* variants have typically been associated with NDD with brain anomalies and dysmorphic facies (NEDBAF, MIM #618577). Conversely, our patient presented with an unusually severe and multisystem phenotype, featuring brain malformations, widespread organomegaly, and congenital anomalies of the hands, including bilateral contractures. These findings suggest that the p.(T17R) variant may be associated with a distinct and more severe *RAC3*-related phenotype, possibly driven by variant-specific pathogenic mechanisms.

To explore the pathogenic mechanism of the p.(T17R) variant in detail, we conducted in vitro analyses. Biochemical assays demonstrated markedly enhanced GTP/GDP exchange activity in the mutant, with a preferential exchange for GDP. GTP hydrolysis activity was undetectable in our assay conditions, likely due to stabilization in the GTP-bound state. Interestingly, while overexpression of RAC3-T17R in primary hippocampal neurons impaired neuronal differentiation—consistent with the behavior of an active, GTP-bound form—cells expressing this variant seldom displayed the round morphology with lamellipodia that is characteristic of RAC activation. Interestingly, another disease-related *RAC3* variant located within the G1 box, the p.(G12R), has been shown to be an active form in vitro [8], although its in vivo functional properties are currently uncharacterized. The clinical phenotype associated with the p.(G12R) variant was considerably milder compared to our patient, with mild extra-neurological abnormalities essentially limited to arachnodactyly and pes planus. Further studies using appropriate model systems will be essential to dissect the mechanistic basis underlying the divergent clinical phenotypes associated with these G1 box variants.

Another *RAC3* variant located in the Switch II region, p.(R66W), has also been associated with severe systemic features, including clenched hands, overlapping fingers, and congenital talipes equinovarus [9]. However, the p.(T17R) and p.(R66W) variants perturb downstream signaling pathways in distinct ways: RAC3-R66W acts as an active form, capable of interacting with downstream effectors such as PAK1, MLK2, and N-WASP [12], whereas RAC3-T17R showed only minimal interaction with these effectors. Interestingly, despite affecting downstream signaling in distinct ways, both RAC3-T17R and RAC3-R66W fail to activate SRF-, AP1-, and NFκB-mediated transcription, suggesting a shared impairment in these transcriptional pathways. In vivo analysis using in utero electroporation demonstrated that acute expression of RAC3-R66W and RAC3-T17R resulted in defective cortical neuron migration during corticogenesis, with a more severe phenotype observed in RAC3-R66W-expressing neurons. These findings collectively suggest that the two variants differentially perturb downstream signaling pathways in a variant-specific and spatiotemporally regulated manner, although with some overlapping outcomes. These functional signatures may play a significant role in determining the overlapping but distinct clinical phenotypes observed in human patients.

Although *RAC3* is predominantly expressed in the brain, the severe extra-neurological anomalies observed in our case and in the subject harboring the p.(R66W) variant suggest that RAC3 also plays essential roles in systemic development. This notion is supported by gene expression data showing detectable *RAC3* expression in various peripheral tissues (https://www.genecards.org/cgi-bin/carddisp.pl?gene=RAC3&keywords=RAC3, accessed on 1 June 2025). RAC1 shares 93% amino acid sequence identity with RAC3. De novo variants in RAC1 cause a broad spectrum of NDDs, predominantly characterized by intellectual disability, global developmental delay, and brain size anomalies (secondary microcephaly or macrocephaly) [29]. Although *RAC1* is ubiquitously expressed across various tissues, the clinical manifestations associated with RAC1 variants are predominantly neurodevelopmental, and systemic manifestations are uncommon. Indeed, extra-neurological abnormalities, particularly involving the cardiovascular system, have been only described in some cases [29]. However, severe systemic malformations resulting in early postnatal death have been recently reported in a single subject harboring the p.(Y40H) variant in *RAC1*. This patient presented with VACTERL association—a congenital condition characterized by malformations of the vertebrae, anus, heart, kidneys, trachea, esophagus, and limbs [30].

The systemic phenotype associated with the p.(Y40H) variant in RAC1 bears some resemblance to the fetopathy observed in our case. This similarity between the RAC1 p.(Y40H) and RAC3 p.(T17R) variants raises the possibility that both substitutions perturb overlapping or common intracellular signaling pathways, contributing to systemic developmental defects beyond the nervous system. Given the functional similarity and high sequence homology between RAC1 and RAC3, it is plausible that specific, structurally critical variants—such as RAC1 p.(Y40H) and RAC3 p.(T17R)—may converge on a shared pathogenic mechanism. These structural perturbations can result in severe and multisystem developmental defects. Therefore, the p.(Y40H) and p.(T17R) variants might represent unique examples of the variable expressivity of RAC dysfunction within the spectrum of *RAC1*- and *RAC3*-related disorders. In this scenario, it is possible that only particular variants, affecting critical functional residues, can override tissue-specific regulatory mechanisms and disrupt broader organogenesis processes.

Although RAC3-T17R has a higher affinity for GDP than for GTP, this variant may still exist in a GTP-bound form, given that intracellular GTP concentrations are much higher than GDP. If this is the case, the minimal affinity of the variant for well-characterized RAC effectors such as PAK1, MLK2, and N-WASP in pull-down assays (Figure 6) raises the possibility that RAC3-T17R aberrantly interacts with as-yet-unidentified effectors, leading to inappropriate hyperactivation of specific downstream signaling pathways. Conversely, if RAC3-T17R is preferentially stabilized in an inactive GDP-bound state, the loss of interaction with canonical RAC effectors may exert a dominant-negative effect by sequestering upstream regulators, thereby preventing the transmission of signals to downstream effectors. Given that more than 30 RAC effectors have been identified to date, it is plausible that RAC3-T17R perturbs a broader network of downstream signaling pathways, ultimately resulting in the complex and systemic phenotype observed in our patient. Future studies will be essential to elucidate the full spectrum of effectors and signaling pathways impacted by this and potentially similar *RAC3* variants, thereby contributing to a more comprehensive delineation of human disease pathophysiology.

In summary, our findings not only expand the clinical and molecular spectrum of *RAC3*-related disorders but also highlight that specific *RAC3* variants may exert peculiar and severe pathogenic effects beyond previously recognized neurodevelopmental phenotypes. These results emphasize the importance of considering *RAC3* variants in the differential diagnosis of congenital malformation syndromes, particularly in prenatal settings where multisystem anomalies are detected. The identification of additional cases and the detailed functional analysis of novel variants in model systems will be essential to elucidate the tissue-specific roles of RAC3 and the variant-specific mechanisms underlying complex clinical presentations.

## Figures and Tables

**Figure 1 cells-14-01499-f001:**
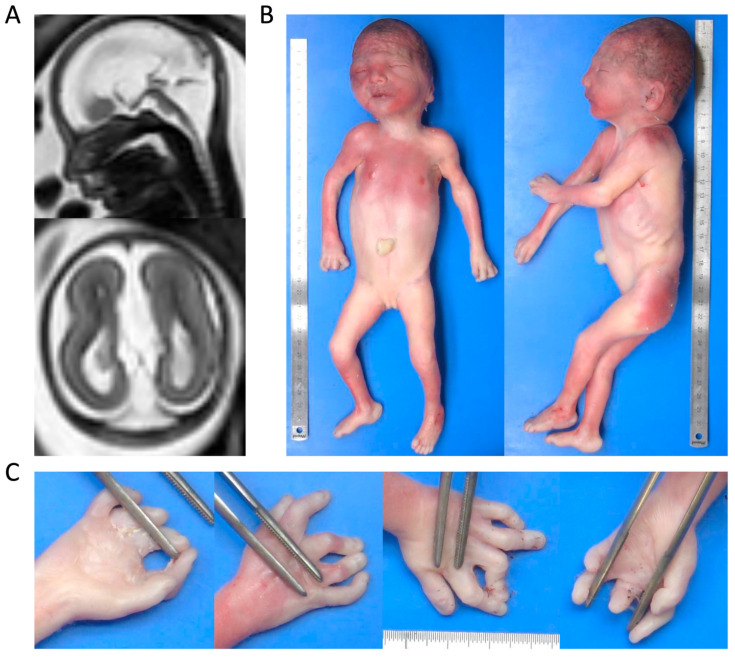
Post-mortem clinical and imaging findings. (**A**) Prenatal brain MRI at 23 weeks’ gestation. Sagittal and axial T2-weighted MRI scans reveal white matter thinning with corpus callosum agenesis, accompanied by secondary septal defects. The lateral ventricles are enlarged and abnormally shaped. There is hypoplasia of the cerebellar vermis, resulting in enlargement of the cisterna magna, along with dysgenesis of the brain characterized by an elongated pons. (**B**) Autoptic findings. The fetus exhibits microcephaly and distinct dysmorphic features, including a receding forehead, large mouth with thin lips, an ogival palate, a short lingual frenulum, and large, malformed ears. Additionally, organomegaly is present, affecting the lungs, heart, kidneys, adrenal glands, and spleen. (**C**) Hand abnormalities. The fetus presents with clenched fists and flexion contractures involving the fingers.

**Figure 2 cells-14-01499-f002:**
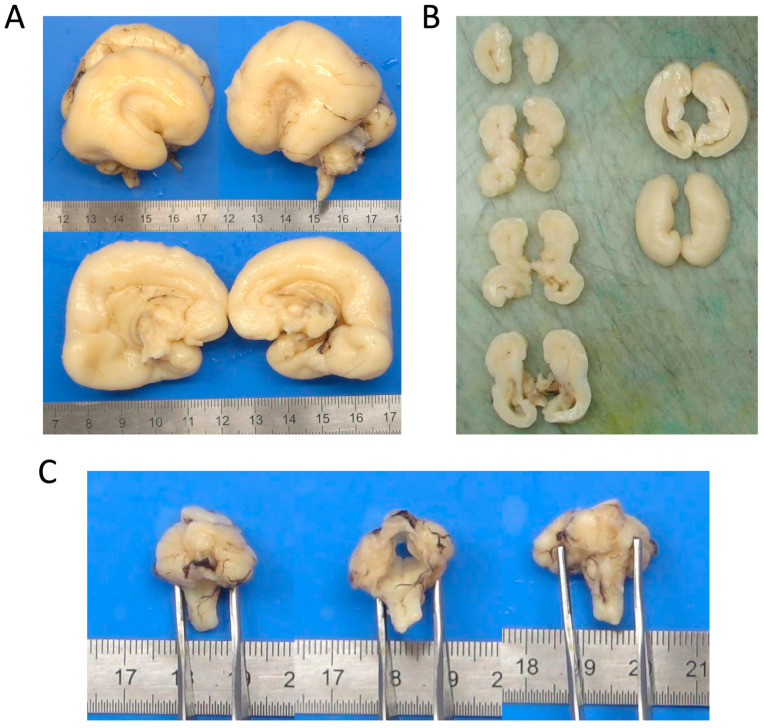
Post-mortem brain examination. (**A**) Sagittal section of the fetal brain. The brain is small, with complete agenesis of the corpus callosum, accompanied by a lack of opercularization of the sylvian fissures. (**B**) Axial section of the hemispheres. There is likely involvement of the white matter, with unusually sharp distinction between the cortical surface and the subcortical white matter, suggesting an underlying neuronal migration abnormality. (**C**) Detailed view of the brainstem. There is severe hypoplasia of the cerebellar vermis, accompanied by a small and dysmorphic pons.

**Figure 3 cells-14-01499-f003:**
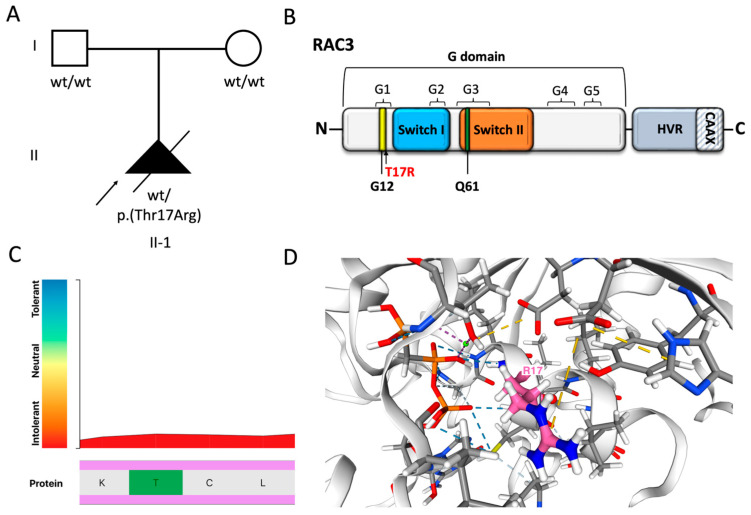
Genetic findings. (**A**) Family pedigree. The proband was conceived by unrelated, healthy parents and was found to harbor a de novo variant in *RAC3*, NM_005052.3: c.50C>G p.(T17R) (*WT* = wild type; *Mut* = mutation). (**B**) Schematic illustration of RAC3 GTPase and mapping of the p.(T17R) variant. The G domain contains the Switch I and II regions and G1-G5 boxes, and includes highly conserved residues such as G12 and Q61. The C-terminal hypervariable region (HVR) contains the CAAX box, a consensus sequence necessary for posttranslational modifications that regulate subcellular localization of the protein. The p.(T17R) substitution affects the last residue of the G1 box and is located near the Switch I domain. (**C**) Intolerance analysis according to Metadome (https://stuart.radboudumc.nl/metadome/dashboard, accessed on 1 July 2025). Thr17 is highly intolerant to genetic variation (score = 0.15) and is located within a functionally constrained region of the protein, suggesting that alterations at this site are likely pathogenic. (**D**) The p.(T17R) substitution is predicted to destabilize the protein structure, with an estimated ΔΔG of 3 kcal/mol. This residue is located 2.0 Å from the binding site for phosphomethylphosphonic acid-guanylate ester (GCP) and within the nucleotide phosphate-binding region for GTP, likely impacting these interactions. Furthermore, the substitution of Thr17 with Arg, which has a different shape and charge, is predicted to alter hydrogen bonding (blue and light blue) and ionic interactions (yellow) with neighboring residues. Predictions made according to Michelanglo (https://michelanglo.sgc.ox.ac.uk/venus, accessed on 1 July 2025).

**Figure 4 cells-14-01499-f004:**
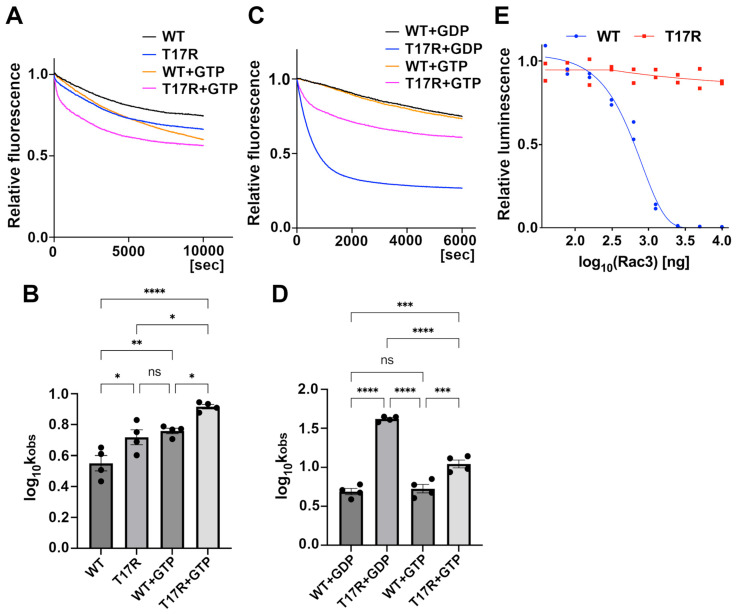
Impact of the p.T17R variant on the affinity of RAC3 proteins for GDP and GTP, and on GTP hydrolysis activity. (**A**–**D**) Mant-GDP-dissociation assay. (**A**) Fluorescently labeled mant-GDP-loaded His-tagged RAC3 (WT) and RAC3-T17R (T17R) were incubated with or without a non-hydrolysable GTP analog, GppNHp, and fluorescence changes were monitored over time to assess exchange activity. (**B**) The dissociation kinetics of mant-GDP from WT and T17R were evaluated, and the rate constants (k_obs_ [×10^−5^/s]) were calculated based on the data from panel *A*. The data were analyzed using one-way ANOVA with Tukey’s post hoc test and shown as interleaved scatter with bars (*p* < 0.05). Sample sizes: WT, *n* = 4; T17R, *n* = 4; WT + GTP, *n* = 4; T17R + GTP; *n* = 4. WT vs. T17R, *p* = 0.031; WT vs. WT + GTP, *p* = 0.0078; WT vs. T17R + GTP, *p* < 0.0001; T17R vs. WT + GTP, *p* = 0.86, T17R vs. T17R + GTP, *p* = 0.011; WT + GTP vs. T17R + GTP, *p* = 0.043. ns: not significant, * *p* < 0.05, ** *p* < 0.01, **** *p* < 0.0001. (**C**) Mant-GDP-loaded WT and T17R were incubated with GDP or GppNHp, and fluorescence changes were then monitored over time. (**D**) The dissociation kinetics of mant-GDP from WT and T17R were evaluated, and the rate constants (k_obs_ [×10^−5^/s]) were calculated based on the data from panel *C*. The data were analyzed using one-way ANOVA with Tukey’s post hoc test and shown with interleaved scatter with bars (*p* < 0.05). Samples sizes: WT, *n* = 4; T17R, *n* = 4; WT + GTP, *n* = 4; T17R + GTP; *n* = 4. WT + GDP vs. T17R + GDP, *p* < 0.0001; WT + GDP vs. WT + GTP, 0.92; WT + GDP vs. T17R + GTP, *p* = 0.0004; T17R + GDP vs. WT + GTP, *p* < 0.0001, T17R + GDP vs. T17R + GTP, *p* < 0.0001; WT + GTP vs. T17R + GTP, *p* = 0.001. ns: not significant, *** *p* < 0.001, **** *p* < 0.0001. (**E**) GTP hydrolysis assay. GTPase activity of WT and T17R was measured using the GTPase-Glo assay kit, which quantifies changes in GTP concentration. Sample sizes: WT, *n* = 3; T17R, *n* = 3. The sigmoidal fitting curve could not be generated for the T17R variant, preventing the determination of the half-maximal effective concentration (EC50). Statistical comparisons were not performed.

**Figure 5 cells-14-01499-f005:**
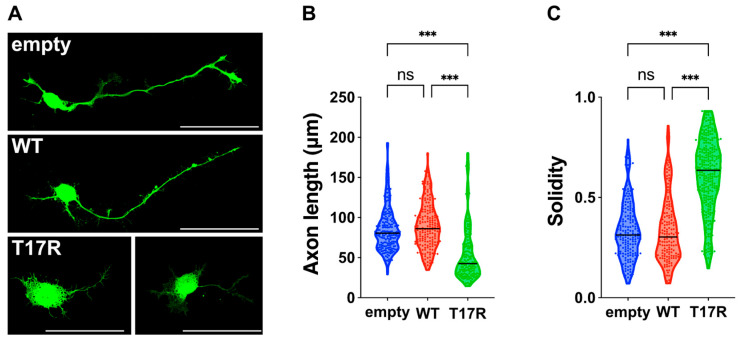
Effects of the p.(T17R) variant on neuron morphology in vitro. (**A**) Primary hippocampal neurons were isolated from E16 embryos and electroporated with pCAG-EGFP (0.1 μg) along with either pCAG-Myc (empty), pCAG-Myc-RAC3 (WT), or pCAG-Myc-RAC3-T17R (0.3 μg each). After 3 days in culture, neurons were fixed and immunostained using anti-GFP (green). Scale bars, 50 μm. (**B**,**C**) Morphometric analysis of neurons shown in (**A**). (**B**) Axon length, defined as the longest neurite in each GFP-labeled neuron, was quantified and presented as violin plots with dots. Neuron counts: empty, *n* = 196; WT, *n* = 197; T17R, *n* = 160. Statistical significance between vector (empty), WT, and T17R was determined using one-way ANOVA with Tukey’s post hoc test (*p* < 0.05). empty vs. WT, *p* = 0.3; empty vs. T17R, *p* < 0.001; WT vs. T17R, *p* < 0.001. (**C**) Cell solidity was measured in GFP-positive neurons and shown as violin plots with boxplots. “Solidity” is the ratio of the area of a cell to the area of a convex hull of the cell [16]. Neuron counts: empty, *n* = 181; WT, *n* = 157; T17R, *n* = 290. Statistical significance between vector (empty), WT, and T17R was determined using one-way ANOVA with Tukey’s post hoc test (*p* < 0.05). empty vs. WT, *p* = 0.97; empty vs. T17R, *p* < 0.001; WT vs. T17R, *p* < 0.001. ns: not significant, *** *p* < 0.001.

**Figure 6 cells-14-01499-f006:**
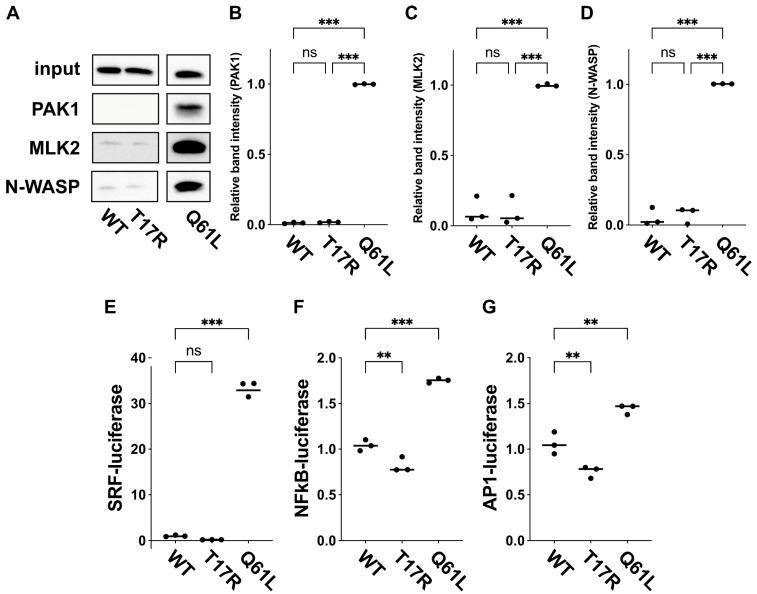
Effects of the p.(T17R) variant on interaction with downstream effectors. (**A**) Evaluation of binding to the RBRs of PAK1, MLK2, and N-WASP. COS7 cells were transfected with pCAG-Myc-RAC3 (WT), -RAC3-T17R, or -RAC3-Q61L (0.3 μg each). GST pull-down assays were conducted using GST-tagged RBRs from PAK1, MLK2, and N-WASP (5 μg each) as in “Materials and methods”. Bound RAC3 proteins were detected via Western blot using anti-Myc. Input samples were also analyzed to verify equivalent expression levels. Full Western blot data of this figure and blot data with molecular marker bands are provided in Appendix A, respectively. (**B**–**D**) The intensity of RAC3 band bound to the GST-fused RBR of PAK1 (**B**), MLK2 (**C**), or N-WASP (**D**) was quantified. The relative band intensity is displayed, with RAC3-Q61L set at 1.0. Samples sizes: *n* = 3 replicates. Statistical significance between WT and each variant was assessed using one-way ANOVA with Tukey’s post hoc test (*p* < 0.05). (B) WT vs. T17R, *p* = 0.32; WT vs. Q61L, *p* < 0.001; T17R vs. Q61L, *p* < 0.001. (C) WT vs. T17R, *p* = 0.99; WT vs. Q61L, *p* < 0.001; T17R vs. Q61L, *p* < 0.001. (D) WT vs. T17R, *p* = 0.86; WT vs. Q61L, *p* < 0.001; T17R vs. Q61L, *p* < 0.001. ns: not significant, *** *p* < 0.001. (**E**–**G**) Effects of p.(T17R) on SRF-, NFkB- and AP1-dependent gene transcription. COS7 cells were transfected with pCAG-Myc-RAC3 (WT) or -RAC3-T17R (0.1 μg each) along with luciferase reporter constructs for SRF (**E**), NFkB (**F**), or AP1 (**G**) (0.05 μg each). Reporter activity was measured, normalized to WT (set at 1.0), and presented as scatter plots with bars. Samples sizes: *n* = 3 replicates. Statistical significance was determined using one-way ANOVA with Dunnett’s post hoc test (*p* < 0.05). (E) WT vs. T17R, *p* = 0.52; WT vs. Q61L, *p* < 0.001. (F) WT vs. T17R, *p* = 0.008; WT vs. Q61L, *p* < 0.001. (G) WT vs. T17R, *p* = 0.008; WT vs. Q61L, *p* = 0.003. ns: not significant, ** *p* < 0.01, *** *p* < 0.001.

**Figure 7 cells-14-01499-f007:**
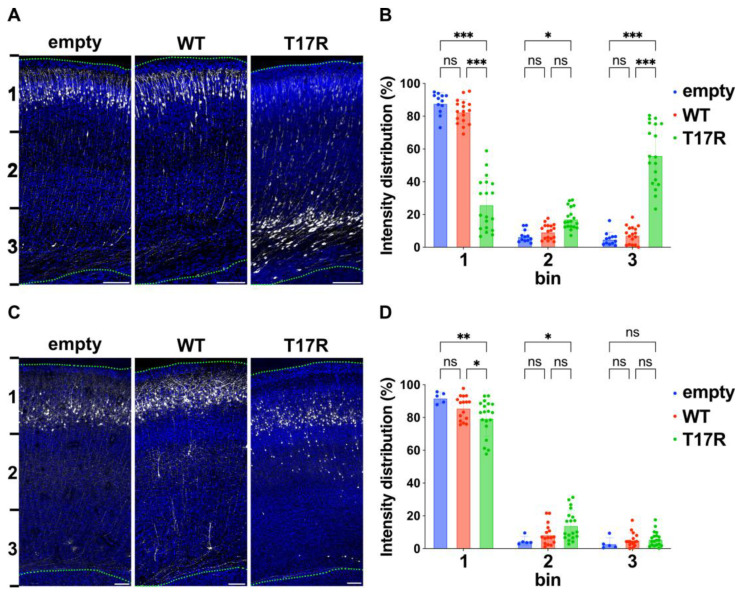
Effects of the p.(T17R) variant on excitatory neuron migration during corticogenesis. (**A**,**C**) Representative images of migration defects of neurons expressing the p.(T17R) variant. E14.5 VZ progenitor cells were electroporated in utero with pCAG-EGFP (0.5 μg) along with either pCAG-Myc (empty), pCAG-Myc-RAC3 (WT), or pCAG-RAC3-T17R (0.1 μg each). Coronal sections were collected at P0 (**A**) and P7 (**C**), and stained with anti-GFP (white) and DAPI (blue). Scale bars, 100 μm. (**B**,**D**) Quantification of (**A**,**C**). Number of replicates, *n* ≥ 5. Statistical significance between WT and the variant was determined using two-way ANOVA with Tukey’s post hoc test and shown with interleaved scatter with bars (*p* < 0.05). (**B**) bin 1: empty vs. WT, *p* = 0.36; empty vs. T17R, *p* < 0.001; WT vs. T17R, *p* < 0.001. bin 2: empty vs. WT, *p* = 0.76; empty vs. T17R, *p* = 0.01; WT vs. T17R, *p* = 0.06. bin 3: empty vs. WT, *p* = 0.80; empty vs. T17R, *p* < 0.001; WT vs. T17R, *p* < 0.001. (**D**) bin 1: empty vs. WT, *p* = 0.23; empty vs. T17R, *p* = 0.003; WT vs. T17R, *p* = 0.04. bin 2: empty vs. WT, *p* = 0.55; empty vs. T17R, *p* = 0.03; WT vs. T17R, *p* = 0.06. bin 3: empty vs. WT, *p* = 0.82; empty vs. T17R, *p* = 0.73; WT vs. T17R, *p* = 0.98. ns: not significant, * *p* < 0.05, ** *p* < 0.01, *** *p* < 0.001.

**Figure 8 cells-14-01499-f008:**
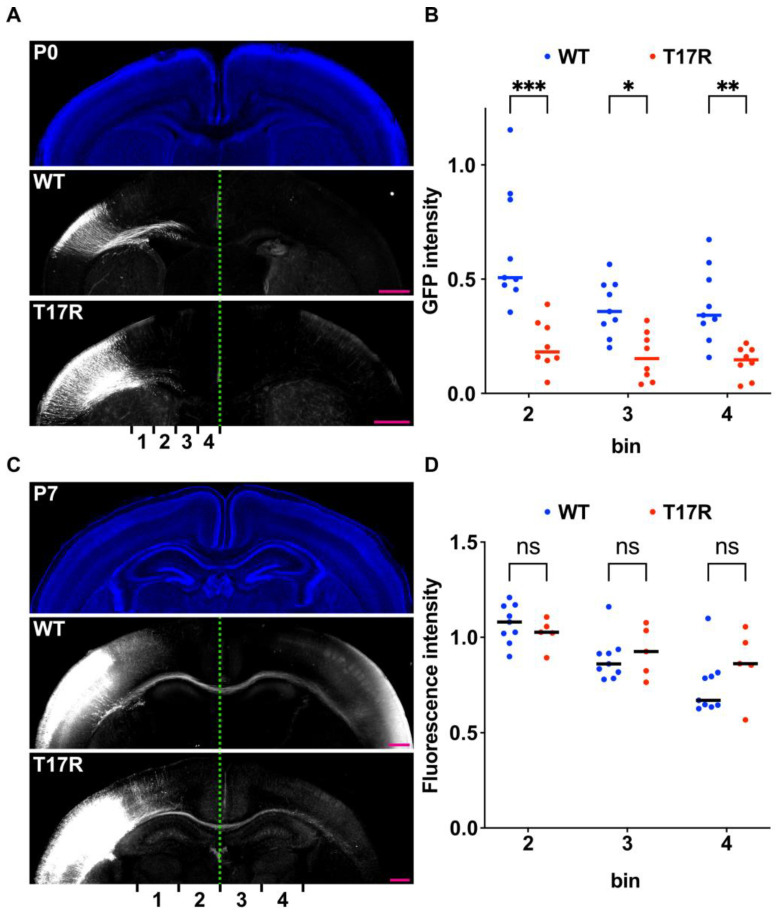
Effects of the p.(T17R) variant on axonal extension during corticogenesis in vivo. (**A**,**C**) pCAG-GFP was co-electroporated at E14 with either pCAG-Myc-RAC3 (WT) or -RAC3-T17R (0.1 μg each). Coronal sections were obtained at P0 (**A**) or P7 (**C**), and GFP fluorescence was used to visualize callosal axons (white). Nuclei were stained with DAPI (blue). Scale bars, 500 μm. (**B**,**D**) Intensity of GFP-labeled callosal axons was quantified at P0 (**B**) and P7 (**D**) across different cortical regions (bins 1–4). Intensities of bins were normalized with bin 1 as 1.0. Statistical significance was assessed using two-way ANOVA with Šídák’s post hoc test and shown with interleaved scatter plots (*p* < 0.05). (**B**) The number of brains analyzed was as follows: WT, *n* = 9; T17R, *n* = 8. bin 2, *p* < 0.001; bin 3, *p* = 0.02; bin 4, *p* = 0.005. (**D**) The number of brains was as follows: WT, *n* = 9; T17R, *n* = 5. bin 2, *p* = 0.86; bin 3, *p* = 0.95; bin 4, *p* = 0.32. ns: not significant, * *p* < 0.05, ** *p* < 0.01, *** *p* < 0.001.

**Figure 9 cells-14-01499-f009:**
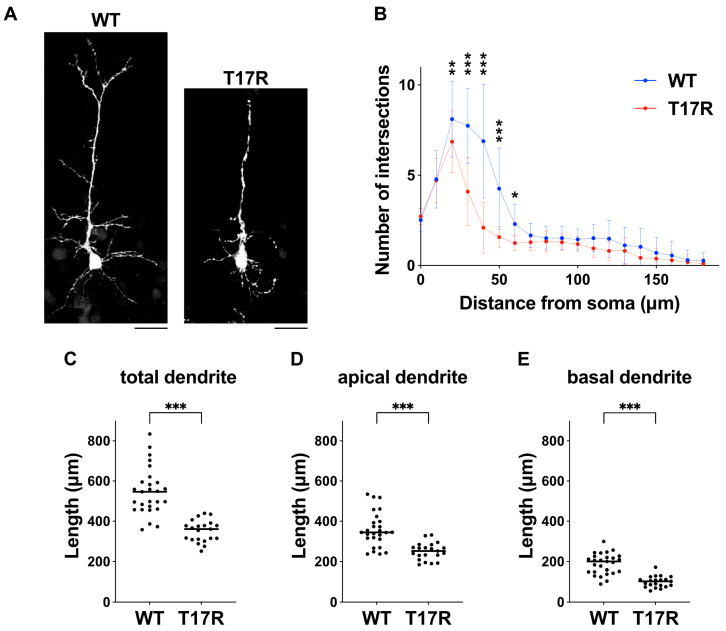
Effects of the p.(T17R) variant on dendritic growth in vivo. (**A**) pCAG-GFP was co-electroporated at E14 with either pCAG-Myc-RAC3 (WT) or pCAG-Myc-RAC3-T17R (0.1 μg each). Coronal sections were obtained at P7, and GFP fluorescence was used to visualize callosal axons (white). Representative average Z-stack projection images of GFP-labeled cortical neurons in the upper cortical plate are shown. Scale bars, 30 μm. (**B**) Dendritic branching in (**A**) was quantified using Sholl analysis. Six brains were analyzed per condition. Number of replicates: *n* ≥ 21. Error bars represent SD. Statistical significance was assessed using two-way ANOVA followed by Šídák’s post hoc test. * *p* < 0.05, ** *p* < 0.01, *** *p* < 0.001. (**C**–**E**) Dendritic length measurements of neurons shown in (**A**) are presented as scatter plots for (**C**) total, (**D**) apical, and (**E**) basal dendrites. Statistical significance was calculated using unpaired *t* test (*p* < 0.05). (**C**) *** *p* < 0.001. (**D**) *** *p* < 0.001. (**E**) *** *p* < 0.001.

## Data Availability

The data used to support the conclusions in this paper are available from the corresponding author upon reasonable request. Genomic data included in the study are available in LOVD at https://databases.lovd.nl/shared/variants/RAC3 (accessed on 1 July 2025).

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
