# Peer review of "A Pleiotropic and Functionally Divergent RAC3 Variant Disrupts Neurodevelopment and Impacts Organogenesis"

_cells, 2025, doi:10.3390/cells14191499_

Round 1

Reviewer 1 Report

Comments and Suggestions for Authors

The manuscript titled “A pleiotropic and functionally divergent RAC3 variant disrupts 2 neurodevelopment and impacts organogenesis” reported a case where a fetus with severe congenital anomalies contains a de novo Rac3 missense T17R. The authors also characterized the biochemical and the biological properties of the mutant. Rac3(T17R) is shown to be a typical dominant-negative mutant and change neuron morphology and development both in vitro and in vivo. This study is highly valuable for documenting and understanding the mutagenetic effects and mechanisms of Rac3. Publication after revision is recommended without a doubt.

However, there is one major flaw in the biochemical study of Rac3 T17R mutants. T17 is well characterized in Rac and Ras GTPases. Both T17N and T17V mutations reduce Rac1 GTP binding affinity by at least 10 folds (PMID: 1643658). Figure 4A shows that the GDP dissociation for T17R is roughly 10 times faster than WT. The observation does not necessarily indicate the T17R is locked in a GTP-bound state but can also be explained by a decrease in GTP binding affinity of T17R that results in more nucleotide-free Rac3. Control experiments are required where fluorescently labeled mantGDP-loaded Rac3(WT) and Rac3(T17R) were incubated without non-hydrolysable GTP analog. Depending on the results Rac3(T17R) may function as a nucleotide-free mutant instead of “an activated variant” as concluded in the manuscript. I think the former explanation is more likely to be the case.

Author Response

Comment 1: However, there is one major flaw in the biochemical study of Rac3 T17R mutants. T17 is well characterized in Rac and Ras GTPases. Both T17N and T17V mutations reduce Rac1 GTP binding affinity by at least 10 folds (PMID: 1643658). Figure 4A shows that the GDP dissociation for T17R is roughly 10 times faster than WT. The observation does not necessarily indicate the T17R is locked in a GTP-bound state but can also be explained by a decrease in GTP binding affinity of T17R that results in more nucleotide-free Rac3. Control experiments are required where fluorescently labeled mantGDP-loaded Rac3(WT) and Rac3(T17R) were incubated without non-hydrolysable GTP analog. Depending on the results Rac3(T17R) may function as a nucleotide-free mutant instead of “an activated variant” as concluded in the manuscript. I think the former explanation is more likely to be the case.

Response: We appreciate the reviewer’s insightful comment. To clarify whether RAC3-T17R is indeed locked in a GTP-bound state or instead exists predominantly in a nucleotide-free form, we conducted a control experiment in which mantGDP-loaded wild-type RAC3 and RAC3-T17R were incubated in the absence of a non-hydrolysable GTP analog. The results showed that mantGDP dissociated from RAC3-T17R slightly more rapidly than from RAC3-WT. In the case of wild-type RAC3, the presence of non-hydrolysable GTP slightly accelerated the release of mantGDP, likely reflecting replacement of GDP with GTP. In contrast, in the presence of GTP, mantGDP dissociation from RAC3-T17R was markedly enhanced, suggesting active nucleotide exchange.

Importantly, when we attempted to evaluate the GTP/GDP exchange activity of RAC3-T17N, we were unable to detect mantGDP binding, suggesting that this mutant adopts a nucleotide-free conformation with poor affinity for guanine nucleotides.

Taken together, these findings may imply that the guanine nucleotide-binding pocket of RAC3-T17R remains intact and capable of interacting with both GTP and GDP. While this supports the interpretation that RAC3-T17R may function as an active, GTP-loaded variant, we cannot exclude the possibility that RAC3-T17R is predominantly in a nucleotide-free state with low affinity for both GTP and GDP. We described these results in the revised manuscript (lines 285 - 307).

In light of this, we have revised the manuscript to tone down the previous assertion in the Abstract section and now acknowledge the possibility that RAC3-T17R may exist in a nucleotide-free conformation (lines 302 - 303). We hope the reviewer finds this clarification acceptable.

Reviewer 2 Report

Comments and Suggestions for Authors

The authors Sugawara et al. identified a new RAC3 variant affecting neurodevelopment and organogenesis. The findings are novel and important. The strength of this paper is in combination of postmortem, in silico, in vitro and in vivo approach and in the multiple assays used to investigate step by step the effects of RAC3-T17R variant. However, there are some points that require modification, especially regarding methodology, the statistical analysis of the data and its presentation. 

Abstract: 

1. The study design is unclear. Please state what experiments were conducted in silico, in vitro or using animal model. 

Materials and methods: 

2. Regarding complexity of performed studies the paper would benefit from addition of schematic representation of experimental design depicting the order of testing procedures. 

3. Please provide full name and catalogue numbers of the main chemical reagents, antibodies and kits used in the study. 

4. Please provide the source and ATCC code for the commercial cell line. 

5. Some pertinent information is missing about animals (age, number etc.) used in the study and the housing conditions. Please provide all necessary information according to ARRIVE checklist. 

6. Why did authors choose to perform microinjections at E14? Please describe what neurodevelopmental processes occur within the chosen period? Instead of “designated postnatal days” (line 158) please provide the exact postnatal days along with rationale for the choice. 

7. The Western blot analysis is only superficially provided in the caption of Fig. 6.  Please provide a detailed description of how the procedure was conducted. The protein ladder in the supplementary material is not visible. 

8. Statistical analysis: the description of statistical methods used in the study is insufficient. Please specify the types of variance analysis used in the study. The authors mention only that Dunnett’s and Tukey’s test were used (please add that these are post hoc tests), while Sidak’s post hoc test was applied (Fig. 8). The authors declare conducting analysis using Prism 9 software, while just above (line 180) also usage of R software is mentioned. 

Results: 

9 Fig. 4B: The authors used an unpaired t-test that is a parametric test. Why is the data then presented as box and whiskers plot, which is appropriate for non-parametric tests? What is depicted by the horizontal line, box frame, and whiskers?  

Author Response

Comment 1: The study design is unclear. Please state what experiments were conducted in silico, in vitro or using animal model. 
Response: We appreciate the reviewer’s comment. As suggested by the reviewer, we have revised the Abstract to clearly state our study design and the experimental tiers employed. Specifically, we now indicate that we performed in silico variant prioritization and structural/pathogenicity predictions, in vitro biochemical and cell‑based assays (COS7 cells and primary hippocampal neurons), and in vivo analyses using a mouse in utero electroporation model to interrogate neuronal migration, axon growth, and dendritic development.

Materials and methods: 

Comment 2: Regarding complexity of performed studies the paper would benefit from addition of schematic representation of experimental design depicting the order of testing procedures. 
Response: We appreciate the reviewer’s insightful comment. In response, we added a schematic representation of experimental design depicting the order of testing procedures in the revised manuscript as Supplementary Fig. S1. 

Comment 3: Please provide full name and catalogue numbers of the main chemical reagents, antibodies and kits used in the study. 
Response: Following the reviewer's suggestion, we have provided full name and catalogue numbers of the main chemical reagents, antibodies and kits used in the study in the “Materials and Methods” section in the revised manuscript. 

Comment 4: Please provide the source and ATCC code for the commercial cell line. 
Response: Following the reviewer’s suggestion, we have added the source and ATCC code for the COS7 cell line in the revised manuscript (lines 148–149).

Comment 5: Some pertinent information is missing about animals (age, number etc.) used in the study and the housing conditions. Please provide all necessary information according to ARRIVE checklist. 
Response: We agree with the reviewer’s comment. We have now provided all relevant information regarding animal age, number, and housing conditions for the in utero electroporation experiments, in accordance with the ARRIVE checklist (lines 181–186 in the revised manuscript).

Comment 6: Why did authors choose to perform microinjections at E14? Please describe what neurodevelopmental processes occur within the chosen period? Instead of “designated postnatal days” (line 158) please provide the exact postnatal days along with rationale for the choice. 
Response: We thank the reviewer for this comment. Neural progenitors in the ventricular zone (VZ) at E14 are known to give rise to layer II/III pyramidal neurons in the mature cortex. Because these neurons are localized in the middle cortical layers and display relatively uniform morphology, this timing is optimal for electroporation to analyze cortical neuron morphology. We have added a sentence to clarify that neuronal progenitor cells at E14.5 in the VZ are destined to become layer II/III pyramidal neurons in the adult brain (lines 178–179 in the revised manuscript).

Regarding the fixation timing, we selected specific time points according to the developmental processes being examined: we fixed brains at E16–P7 for neuronal migration during corticogenesis, at P0–P7 for axon elongation, and at P7–P21 for dendrite development. Following the reviewer’s suggestion, we have now provided the exact postnatal days of fixation in the “Materials and Methods” section (lines 180–181 in the revised manuscript).

Comment 7: The Western blot analysis is only superficially provided in the caption of Fig. 6.  Please provide a detailed description of how the procedure was conducted. The protein ladder in the supplementary material is not visible. 
Response: We thank the reviewer for this comment. We have revised the “Materials and Methods” section to provide a detailed description of the Western blot and pull-down procedures (lines 154–168 in the revised manuscript). In addition, we have expanded the Results section for Fig. 6 to more clearly explain the experimental setup and findings (lines 354–358). We thank the reviewer for this comment.

The protein ladder is not visible in the supplementary Western blot images because the detection of protein bands was performed using a fluorescence-based imaging system. The molecular weight markers we used are pre-stained in blue and can only be visualized under bright-field illumination, whereas the protein bands are detected in the fluorescence channel. Therefore, it is technically impossible to display the fluorescent signals of the target proteins and the pre-stained markers simultaneously in a single image. To ensure clarity, we confirmed the positions of the molecular weight markers during imaging and aligned the fluorescent bands accordingly. We hope that this explanation addresses the reviewer’s concern.

Comment 8: Statistical analysis: the description of statistical methods used in the study is insufficient. Please specify the types of variance analysis used in the study. The authors mention only that Dunnett’s and Tukey’s test were used (please add that these are post hoc tests), while Sidak’s post hoc test was applied (Fig. 8). The authors declare conducting analysis using Prism 9 software, while just above (line 180) also usage of R software is mentioned. 
Response: We appreciate the reviewer’s comments on our statistical analysis section. In the revised manuscript, we have now specified in the the types of variance analyses used (one-way or two-way ANOVA), and clarified that Dunnett’s, Tukey’s, and Sidak’s tests were applied as post hoc tests, depending on the experimental design (lines 210–217). We have also corrected the inconsistency regarding the statistical software. All analyses were performed using GraphPad Prism 9 (GraphPad Software, San Diego, CA), and we have removed the mention of R software to avoid confusion.

Results: 

Comment 9: Fig. 4B: The authors used an unpaired t-test that is a parametric test. Why is the data then presented as box and whiskers plot, which is appropriate for non-parametric tests? What is depicted by the horizontal line, box frame, and whiskers?  
Response: We thank the reviewer for this important observation. As stated in the comment made by another reviewer, we have newly conducted mantGDP assay in the presence or absence of a non-hydrolysable GTP. In the revised manuscript, we need to perform a statistical analysis of four groups, for which and we used a one-way ANOVA with a post-hoc Tukey’s test to analyze the data. In the figure, the horizontal line within the box indicates the median, the box edges represent the 25th and 75th percentiles (interquartile range), and the whiskers extend to the minimum and maximum values. This information has been added to the figure legend in the revised manuscript to avoid confusion.

Round 2

Reviewer 1 Report

Comments and Suggestions for Authors

I am not fully convinced of the result from the biochemical assays.

1). The authors claim that “At the start of the assay, the relative fluorescence intensity 287 of RAC3-T17R was approximately 1.3 times higher than that of wild-type RAC3”, suggesting the GDP binding affinity for T17R mutant is higher than WT. On the contrary, the dissociation of T17R is faster than WT, suggesting a lower affinity.

2). It has been documented that the GDP binding affinity for Rac1 (PMID: 1643658) and H-Ras (PMID: 3145408) is similar for both T17N and WT proteins. Given the sequence similarity of Rac1 and Rac3, it is puzzling that the authors were not able to load mantGDP onto the Rac3 T17N mutant.

For better credibility of the results, the authors should add one control experiment in Fig. 4A: comparing the GDP exchange for both WT and T17R mutant by incubating fluorescently labeled mantGDP-loaded His-tagged RAC3 (WT) and RAC3-T17R with unlabeled GDP using the same concentration as non-hydrolysable GTP analog. If the T17R is indeed more GTP bound than WT type, the GDP exchange kinetics should be similar between T17R and WT.

Author Response

Reviewer #1

Comment 1: I am not fully convinced of the result from the biochemical assays.

The authors claim that “At the start of the assay, the relative fluorescence intensity of RAC3-T17R was approximately 1.3 times higher than that of wild-type RAC3”, suggesting the GDP binding affinity for T17R mutant is higher than WT. On the contrary, the dissociation of T17R is faster than WT, suggesting a lower affinity.

Response 1: In this sentence, we intended only to indicate that RAC3-T17R retains GDP-binding ability. To avoid this misunderstanding, we have revised the relevant sentence in the manuscript (p.8, lines 286–287). 

Comment 2: It has been documented that the GDP binding affinity for Rac1 (PMID: 1643658) and H-Ras (PMID: 3145408) is similar for both T17N and WT proteins. Given the sequence similarity of Rac1 and Rac3, it is puzzling that the authors were not able to load mantGDP onto the Rac3 T17N mutant.

Response 2: We appreciate the reviewer’s comment. Although previous studies have shown that the GDP-binding affinity of Rac1 and H-Ras is similar for both T17N and wild-type proteins, we observed that mantGDP could not be loaded onto the Rac3 T17N mutant under our experimental conditions. At present, we can only report the results as obtained, and the underlying reasons for this difference remain unclear. We would appreciate the reviewer’s understanding regarding this point.

Comment 3: For better credibility of the results, the authors should add one control experiment in Fig. 4A: comparing the GDP exchange for both WT and T17R mutant by incubating fluorescently labeled mantGDP-loaded His-tagged RAC3 (WT) and RAC3-T17R with unlabeled GDP using the same concentration as non-hydrolysable GTP analog. If the T17R is indeed more GTP bound than WT type, the GDP exchange kinetics should be similar between T17R and WT.

Response 3: We appreciate the reviewer’s insightful comment. Following the suggestion, we conducted the assay using GDP instead of non-hydrolysable GTP. Under these conditions, RAC3-T17R exhibited a more rapid decrease in fluorescence than wild-type RAC3, with a rate even faster than that observed in the presence of non-hydrolysable GTP. These results indicate that, while the accelerated decrease in RAC3-T17R fluorescence in the presence of non-hydrolysable GTP or GDP reflects enhanced nucleotide exchange activity, RAC3-T17R shows a preference for GDP over GTP.

Taken together, these findings suggest that RAC3-T17R biochemically behaves as an inactive, GDP-bound variant, distinct from other pathogenic RAC3 variants reported to date. Thus, RAC3-T17R is not necessarily a constitutively activated form, although we cannot exclude the possibility that it may exist in a GTP-bound state under physiological conditions, where GTP concentrations are substantially higher than GDP. Based on these results, we interpret the p.T17R variant as functioning in a dominant-negative manner toward PAK1, MLK2, and N-WASP pathways. Accordingly, we have withdrawn our initial claim that RAC3-T17R represents an activated version, and have revised the manuscript to amend our previous interpretation. We have highlighted the major changes in red throughout the revised manuscript.

Reviewer 2 Report

Comments and Suggestions for Authors

The authors have thoroughly addressed my comments. In my opinion, the manuscript has been improved.

However, the issue with the Fig. 4B remains unresolved. Previously, the authors performed an unpaired t-test, which is a parametric test. Now (as the authors analyzed four groups) the one-way ANOVA test was applied, which is also a parametric test, thus it compares means between the groups not medians. The data should be then presented as a bar plot with mean and SEM (standard error of the mean). The present visualization of one-way ANOVA results as box and whiskers plot along with the added information “the horizontal line within the box indicates the median, the box edges represent the 25th and 75th percentiles (interquartile range), and the whiskers extend to the minimum and maximum values” (lines 314-316) is incorrect. Please provide appropriate Fig. 4B and its description. Please provide Fig. 4 in higher resolution to ensure readability.

Author Response

Reviewer #2

Comment 1: The authors have thoroughly addressed my comments. In my opinion, the manuscript has been improved. However, the issue with the Fig. 4B remains unresolved. Previously, the authors performed an unpaired t-test, which is a parametric test. Now (as the authors analyzed four groups) the one-way ANOVA test was applied, which is also a parametric test, thus it compares means between the groups not medians. The data should be then presented as a bar plot with mean and SEM (standard error of the mean). The present visualization of one-way ANOVA results as box and whiskers plot along with the added information “the horizontal line within the box indicates the median, the box edges represent the 25th and 75th percentiles (interquartile range), and the whiskers extend to the minimum and maximum values” (lines 314-316) is incorrect. Please provide appropriate Fig. 4B and its description. Please provide Fig. 4 in higher resolution to ensure readability.

Response 1: We appreciate the reviewer’s comment. In the revised manuscript, we have presented the data as a bar plot showing the mean ± SEM (standard error of the mean) and have accordingly revised the figure legend. In addition, Fig. 4B has been updated in higher resolution to ensure readability.

Round 3

Reviewer 1 Report

Comments and Suggestions for Authors

The updated manuscript showed complete biochemical characterization of T17R mutant. I have no more critiques. Publication is recommended.

Author Response

Comment 1: The updated manuscript showed complete biochemical characterization of T17R mutant. I have no more critiques. Publication is recommended.

Response: We are deeply grateful for your insightful and constructive review throughout the process. We truly appreciate your positive recommendation for publication.